# Ocular Symptoms Associated with COVID-19 Are Correlated with the Expression Profile of Mouse SARS-CoV-2 Binding Sites

**DOI:** 10.3390/v15020354

**Published:** 2023-01-26

**Authors:** Julien Brechbühl, Flavio Ferreira, Ana Catarina Lopes, Emilly Corset, Noah Gilliand, Marie-Christine Broillet

**Affiliations:** Department of Biomedical Sciences, Faculty of Biology and Medicine, University of Lausanne, Bugnon 27, CH-1011 Lausanne, Switzerland

**Keywords:** COVID-19, SARS-CoV-2, ACE2, TMPRSS2, mouse sensory systems, olfaction, taste, vision

## Abstract

The COVID-19 pandemic has engendered significant scientific efforts in the understanding of its infectious agent SARS-CoV-2 and of its associated symptoms. A peculiar characteristic of this virus lies in its ability to challenge our senses, as its infection can lead to anosmia and ageusia. While ocular symptoms, such as conjunctivitis, optic neuritis or dry eyes, are also reported after viral infection, they have lower frequencies and severities, and their functional development is still elusive. Here, using combined technical approaches based on histological and gene profiling methods, we characterized the expression of SARS-CoV-2 binding sites (*Ace2/Tmprss2*) in the mouse eye. We found that ACE2 was ectopically expressed in subtissular ocular regions, such as in the optic nerve and in the Harderian/intraorbital lacrimal glands. Moreover, we observed an important variation of *Ace2*/*Tmprss2* expression that is not only dependent on the age and sex of the animal, but also highly heterogenous between individuals. Our results thus give new insight into the expression of SARS-CoV-2 binding sites in the mouse eye and propose an interpretation of the human ocular-associated symptoms linked to SARS-CoV-2.

## 1. Introduction

Since its beginning, the worldwide health crisis generated by the Coronavirus disease 2019 (COVID-19) has engendered significant scientific efforts to understand the phenomena linked to the cellular interactions of its infectious agent, the severe acute respiratory syndrome coronavirus 2 (SARS-CoV-2), and its associated symptoms. In addition to the devastating short and long-term effects at the pulmonary and systemic levels [1,2,3], one of the peculiarities of this virus is that it directly affects our senses, which can go as far as the striking cases of total loss of smell (anosmia) and taste (ageusia) [4,5,6]. From a functional point of view, a direct correlation exists between the viral load measured within the affected organ and the appearance of the associated symptoms [7,8]. With the olfactory and gustatory systems both being sentinels of our environment and directly exposed to the volatile particles of SARS-CoV-2, the early appearance of the symptoms of anosmia and ageusia, therefore, seems consistent. Similarly, the accessibility of the eyes to viral aerosols should render them sensitive to viral attacks. Ocular manifestations related to viral infections are, indeed, common, such as for human herpes or influenza viruses, as well as for other emerging viral diseases (e.g., Ebola and Zika viruses) [9]. Interestingly, in the case of the SARS-CoV-2, only rare extreme cases of blindness and sensory damages, such as conjunctivitis, optic neuritis or dry eyes, have been reported, with a low frequency (up to 25% *vs.* 80% for anosmia and ageusia). Moreover, these symptoms appear late in the course of the viral infection, and they are of reduced severity compared to the ones observed in the senses of chemical perception [10,11,12,13]. One of the main reasons for this divergence lies in the interaction of SARS-CoV-2 with its host cells. The viral recognition is indeed directly linked with the expression of the angiotensin-converting enzyme 2 (ACE2) at the level of the cell membrane [14,15]. In addition, the co-expression of cleaving enzymes, such as the transmembrane serine protease 2 (TMPRSS2), as well as the proximity to the systemic network, are factors favoring the efficiency of viral infection; these factors thus allow SARS-CoV-2 to optimize not only its cellular entry, but also the production of new virions leading to inflammatory mechanisms and cytopathic destruction [14,15,16,17,18]. Thus, a primary viral infection in the eyes would require the ocular surface to be a site of protein expression in the viral binding sites (ACE2/TMPRSS2). However, their expression in humans is only sporadic on the ocular surface, such as at the level of the conjunctiva and the cornea, but also in the different retinal layers (Figure 1a) [19,20,21,22,23,24,25], making the interpretation of ocular symptoms still elusive.

Another characteristic of COVID-19 is that, depending on age or sex, and also on the different emerging variants, the sensory symptoms can differ not only in their manifestation, but also in their severity [26,27,28]. In order to be able to study these features and resolve the underlying mechanisms, animal models have been and are currently beneficial. As such, the hamster and the ferret, both naturally compatible hosts for infections of the initial SARS-CoV-2 variants [29,30,31,32,33], as well as the mouse and the humanized mouse models for ACE2 (hACE2), have been used to demonstrate how, at the olfactory level, the virus did not directly target olfactory sensory neurons (OSNs) but rather the so-called sustentacular supporting cells (Figure 1a) [34,35,36,37,38,39]. At the taste level, similar efforts have been carried out successfully, demonstrating the expression of viral binding sites at the level of the tongue epithelia, as well as in the taste pores where sensory cells are located (Figure 1a) [39,40]. The fundamental knowledge on the effects of SARS-CoV-2 has rapidly expanded thanks to the use of these animals, as well as the generation of new transgenic models that are adaptable to the different variants of the virus and are commonly accessible by biology laboratories throughout the world. The results of this research have been confirmed with human observations and have allowed the anticipation of new viral treatments [41,42,43]. Recently, the abilities of SARS-CoV-2 to rapidly evolve and adapt to the environmental pressure have also been demonstrated. Indeed, it has been reported that new emerging subvariants, such as those from omicron, are highly compatible with mice, thus suggesting that they may be, or may become, the new viral reservoir host [44]. The detailed characterization of the mouse experimental model is, therefore, fundamental in the understanding of its viral infectivity and of the sensory symptoms observed, particularly at the ocular level.

In this study, we characterized the expression of the *Ace2*/*Tmprss2* binding sites of SARS-CoV-2 in the mouse eye. We revealed that ACE2 could be mainly and strongly expressed in different ocular regions, such as the optic nerve, the intraorbital lacrimal and the Harderian glands. Corroborating our histological results, we also observed that the gene expression of these viral binding sites was not only dependent on the sex and on the age of the animal, but that it is also highly heterogeneous between individuals. Our observations allow the interpretation of the heterogenicity of the ocular symptoms that are occurring later and ectopically in humans in the course of the viral infection linked to SARS-CoV-2. They also support the use of the mouse model in future SARS-CoV-2-related research.

## 2. Materials and Methods

### 2.1. Animals

In this study, OMP-GFP knockin mice of both sexes from the C57BL/6 (*Mus musculus*; Janvier Labs) genetic background were used [45,46]. In these mice, the reporter green fluorescent protein (GFP) is expressed under the control of the olfactory marker protein (OMP) promoter [45,47], and allows the visualization of all mature olfactory sensory neurons [48] without impacting ACE2 and TMPRSS2 expression [39]. The mice were housed under a 12 h light/dark cycle in the animal facility of the department, at a temperature fixed between 21 to 23 °C. The mice were euthanized by cervical dislocation or by CO_2_ inhalation. The experimental procedures were in accordance with the Swiss legislation and approved by the EXPANIM committee of the Lemanique Animal Facility Network and the veterinary authority of the Canton de Vaud (SCAV).

### 2.2. Sensory Systems Isolation

Dissection procedures were performed in sterile phosphate buffered saline (PBS; 138 mM NaCl, 2.7 mM KCl, 1.76 mM KH_2_PO_4_, and 10 mM Na_2_HPO_4_, pH 7.4). Tools and equipment were disinfected and RNase removing agent (RNaseZAP^TM^; Sigma) treatments were systematically applied. According to their specific olfactory GFP expression [39,49,50], heterozygous (*Gfp*^+/−^) and homozygous (*Gfp*^+/+^) OMP-GFP mice were used to visualize and precisely extract the GFP-positive dorsal part of the main olfactory epithelium (MOE_d_) under a fluorescent stereomicroscope (M165 FC; Leica, Muttenz, Switzerland). The circumvallate papillae (CV) were removed from the tongues and used as a source of taste buds [39]. The eyes with their associated optic nerves and their neighboring tissues, including muscles and Harderian/intraorbital lacrimal glands, were delicately removed from their orbits [51]. The conjunctiva, as well as the extraorbital lacrimal glands, were not collected during the eye dissection process.

### 2.3. Immunohistochemistry

Protein expression profiles were obtained by fluorescent immunostaining procedures [39,46]. Adult mice (from 16 to 20 months) were sacrificed, the skin was removed and the mouse heads were chemically fixated in 4% paraformaldehyde (PAF 4%, pH 7.4) for 24 h at 4 °C. The eyes were then delicately removed and placed in a sucrose gradient solutions for cryopreservation before being cryosectioned (HM 525NX; Thermo Fisher Scientific, MA, USA) at 20 μm [46]. For the MOE_d_ and the CV, sensory epithelia were included in low melting 4% agar [50,52]. Coronal slices of 120 μm were generated with a vibroslicer (VT1200S; Leica) and collected in ice-cold PBS. Based on their general morphology and/or GFP expression (for the MOE_d_), slices were selected under a stereomicroscope (M165 FC; Leica). A similar and indirect immunostaining procedure was then applied for cryo and floating sections [39]. For that, slices of sensory epithelia were blocked for a minimum of 3 h at room temperature in a permeabilization solution containing 5% normal donkey serum (NDS; Jackson ImmunoResearch, Cambridge, UK), supplemented with a 1% non-ionic detergent (Triton^®^ X-100; Fluka, Aubonne, Switzerland). Specific primary antibodies were then used to simultaneously localize ACE2 and CK18 expression [34,35,39]. The primary antibodies used were directed against ACE2 (Goat anti-ACE2; PA5-47488; Invitrogen; 1:40) and CK18 (Rabbit anti-CK18; PA5-14263; Invitrogen; 1:50). The detection of primary antibodies was performed using fluorochrome-conjugated secondary antibodies against goat (Alexa Fluor Plus 647-conjugated, donkey anti-Goat; A32849; Invitrogen; 1:200|FITC-conjugated, donkey anti-Goat; 705-095-147; Jackson ImmunoResearch; 1:200) or rabbit (Cy3-conjugated, donkey anti-Rabbit; 711-165-152; Jackson ImmunoResearch; 1:200). The slices were then rinsed in 1% NDS solution and the nuclei were counterstained with DAPI (Vectashield^®^; H-1200; Vector Labs, Servion, Switzerland). Control experiments were performed in the absence of primary antibodies. Acquisitions were made by confocal microscopy (SP5; Leica) under a calibrated intensity of the lasers, in order to standardize the observation and the comparison of the signals emitted within each sensory tissue investigated [39]. Slice projections were made with a reconstruction software (IMARIS 6.3; Bitplane).

### 2.4. RNA Isolation and Purification

The isolation and purification of the different RNA samples [39] were performed by pooling together 2–3 MOE_d_ (*Gfp*^+/− and +/+^), 2-3 CV and 1 eye from either young or adult mice with a comparable sex ratio (Table 1). According to the manufacturer’s kit (RNeasy^®^ Plus Mini kit; Qiagen, Aarhus, Denmark), sensory tissues were placed in the buffer RLT Plus with β-mercaptoethanol. Tissue homogenization was achieved through high-speed tissue disruption (TissueLyser II; Qiagen) and the gDNA Eliminator spin column was used to remove genomic DNA. RNeasy spin columns were used, and the total RNA was then eluted in 30 μL of RNase-free water. The RNA samples were purified by ice-cold sodium acetate–ethanol solution steps [53], and the RNA concentrations were assessed according to spectrophotometer measurements (NanoDrop 200c; Thermo Scientific). According to the cDNA synthesis kit (PrimeScript^TM^ 1st strand cDNA Synthesis Kit; Takara, Saint-Germain-en-Laye, France), reverse transcription (RT) was completed using 140 ng of RNA and the random hexamers option to obtain a final volume of 20 μL of cDNA [39].

### 2.5. RT-PCR and RT-qPCR

Gene expression profiles were performed according to RT-polymerase chain reaction (RT-PCR) and quantitative RT-PCR (RT-qPCR) experiments [39]. Briefly, 3 μL of cDNA and 800 nM of the following primers (Microsynth AG, Balgach, Switzerland) were used: *Ace2* (forward 5′-CTACAGGCCCTTCAGCAAAG-3′; reverse 5′-TGCCCAGAGCCTAGAGTTGT-3′; product size of 204 bp [39]), *Tmprss2* (forward 5′-ACTGACCTCCTCATGCTGCT-3′; reverse 5′-TGACAGATGTTGAGGCTTGC-3′; product size of 225 bp [39]) and *Gapdh* (forward 5′-AACTTTGGCATTGTGGAAGG-3′; reverse 5′-ACACATTGGGGGTAGGAACA-3′; product size of 223 bp [39]).

RT-PCR amplifications were initiated with the DNA Polymerase (GoTaq^®^ DNA Polymera; Promega, Dübendorf, Switzerland) under a thermocycler (Veriti™; Applied Biosystems, MA, USA), according to 30 cycles (95 °C for 30 s, 55 °C for 30 s and 68 °C for 45 s). Amplica were visualized according to UV illumination on a 2% electrophoresis gel supplemented with ethidium bromide, and their sizes were confirmed according to a DNA ladder (Bench Top 100 bp DNA Ladder; Promega).

Quantitative RT-PCR (RT-qPCR) experiments were performed with a Real-Time PCR system detector (7500 Fast Real-Time PCR System; Applied Biosystems). For each experiment, all the reactions were performed in triplicate with the SYBR^®^ Green enzyme (Fast SYBR Green Master Mix; Applied Biosystems) under the universal fast-PCR cycling conditions. The different C_T_ (cycle thresholds) were compared using a threshold line of 0.15. The RNA fold change was then calculated according to the comparative 2^(−ΔΔCT)^ method [39], normalized to *Gapdh*. A melting curve analysis, using the derivative reporter of the normalized fluorescence (-Rn′) method, as well as the electrophoresis of qPCR amplica on 2% agarose gels, were used as a diagnostic tool for assessing the specificity of the qPCR products [54]. In parallel, purified amplica (extrAXON DNA Clean-up Kit; Axon Lab, Mont-sur-Lausanne, Switzerland) were sequenced by the Sanger method (Fasteris SA, Plan-les-Ouates, Switzerland). The validity of the sample queries was confirmed by the Blastn process [55]. Local alignments from the identified target sequences (*Ace2*, ID: AB053181.1; *Tmprrs2*, ID: AF199362.1; *Gapdh*, ID: GU214026.1) and from our sample queries were performed by using the T-COFFEE algorithm [56].

### 2.6. Statistical Analysis

Statistical analysis, as well as their associated dot-plot graphics, were computed with GraphPad Prism 9.1.1. The values were expressed as mean ± standard error of the mean (SEM). The comparisons were unpaired and performed with the two-tailed Student’s *t*-tests with Welch’s correction, in the case of non-respect to the homoscedasticity (Fisher *F*-tests). In the absence of Normality, assessed by the Shapiro–Wilk test, Mann–Whitney tests were applied. Significance levels are indicated as follows: * for *p* < 0.05; ** for *p* < 0.01; *** for *p* < 0.001; ns for non-significant.

## 3. Results

### 3.1. Expression Profile of ACE2 in the Mouse Eye

To characterize the expression of the viral binding sites in the mouse eye, we focused on the expression of the main membrane receptor implicated in SARS-CoV-2 cellular recognition, the ACE2 [14,15]. To facilitate the precise localization of its expression, we used our previous observations, which demonstrated that ACE2 expression was correlated with the expression of the cellular marker Cytokeratin 18 (CK18) [39]. CK18 is an intermediate filament protein member of the intracytoplasmic cytoskeleton found in epithelial cells and other tissues such as, for example, pulmonary, peripheral nerves or sensory systems [39,57,58]. Moreover, the cellular expression of CK18 has been used to develop transgenic mice, using it as a promoter of hACE2 expression, in order to significantly sensitize these mice to SARS-CoV-2 infections while allowing it to mimic human lungs and sensory-related symptoms [7,36,59,60,61]. Accordingly, we first confirmed the apical ACE2 expression in the CK18-positive sustentacular cells of the MOE_d_ (Figure 1b), as well as in the tongue epithelial cells and at the level of the taste pores where the CK18-positive sensory cells of the taste buds of the CV papillae are located (Figure 1c). We then performed this histological approach on the eyes of the mouse, and we found the expression of CK18 in different ocular regions (Figure 1d–i). Confirming previous reports, we noticed a striking expression of CK18 in the optical nerve and in the retinal pigment epithelium of the retina (RPE; Figure 1d); this was also found in the different secretary glands, such as in the cells composing the Harderian (Figure 1g) and intraorbital lacrimal glands (Figure 1h) [62,63]. Interestingly, we observed that the expression of ACE2 was highly correlated with the expression of CK18, as we found its expression to be mainly restricted to the RPE layer of the retina (Figure 1d), to the lumen of the tubular alveoli of the Harderian glands (Figure 1g), to the lumen of the acini cells composing the intraorbital lacrimal glands [64] (Figure 1h), as well as to the optical nerve (Figure 1d); this confirmed recent evidence of SARS-CoV-2 tropism in humanized mice [61]. Background signals were assessed by negative controls, and they demonstrated the specificity of our CK18/ACE2 stainings (Figure 1f,i). However, we also found the expression of ACE2 in several CK18-negative tissues, such as blood vessels (Figure 1d,e). These results point out that CK18-expressing cells are not the exclusive sites of ACE2 expression [24,25,39].

In summary, our histological results demonstrate and localize the expression of ACE2 receptors in the mouse eye. Contrary to our observations performed on the MOE_d_ (Figure 1b) and CV (Figure 1c), we noticed, in the eye, an apparent heterogenicity in the intensity of the CK18/ACE2 signals under calibrated laser and confocal acquisition procedures. Although the subregional distributions of the stainings were reproducible (N_eye_ = 6 from 5 different mice), two eyes from different mice showed specific and stronger intensities overall. As an illustration, the indicated signals from the Harderian and those from the intraorbital lacrimal glands (Figure 1g vs. Figure 1h) are from different individuals in which the expression of CK18/ACE2 was distinctively different; this suggests a potential interindividual-dependent expression of SARS-CoV-2-associated binding sites.

### 3.2. Gene Expression of Ace2 and Tmprss2 in the Mouse Eye

We next assessed, at the RNA level, the expression of *Ace2* and *Tmprss2* in the mouse eyes (Figure 2). To avoid any potential endogenous genomic DNA contaminations and/or amplifications, we systematically used deoxyribonuclease (DNAse) procedures and previously designed primers that span the exon–exon junction [39]. As a first examination of the gene expression profiles, we confirmed by RT-PCR (Figure 2a) the ocular expression of these SARS-CoV-2-associated binding genes, as well as their expressions in the dorsal part of the main olfactory epithelium (MOE_d_) and the tongue circumvallate papillae (CV). We found, in each investigated sensory tissue, a single PCR product size at the predicted values, confirming our amplification specificities. Interestingly, we observed different intensities of expression (Figure 2a; *n* > 3 independent experiments) in the different sensory tissues. We further quantified these expression differences by RT-qPCR (*n* > 3 independent experiments) and found a significantly lower level of ocular *Ace2* compared to the MOE_d_ and to the taste CV papillae of the tongue (Figure 2b). Interestingly, *Tmprss2* was also expressed significantly less in the eye compared to MOE_d_, whereas its expression was similar to the one found in the taste CV papillae (Figure 2b). These observations support the notion that the eyes are not the main entry site for a primary SARS-CoV-2 infection, unlike the respiratory and chemosensory organs [7,10,11,12,20,23,61]. To validate the specificity of our amplification products, we next analyzed the profiles of the melting curves of our different samples (Appendix A). We measured, for each investigated gene, a single peak of the melting temperature (*Ace2* T_m_: 77.4 ± 0.05 °C; *Tmprss2* T_m_: 80.7 ± 0.04 °C; *Gapdh* T_m_: 83.5 ± 0.04 °C), implying pure and single qPCR amplica for each sample and gene. Furthermore, these results were confirmed by electrophoresis evaluation (Appendix A), as well as by sequence analysis (Appendix A).

Using a representative population of mice (Table 1), we next compared the interindividual expression of *Ace2*/*Tmprss2* (Figure 2c). Confirming our histological results (Figure 1g,h), we reproducibly observed (*n* > 3 independent experiments) an interindividual heterogenicity, particularly for *Ace2* (Figure 2c). Indeed, the level of *Ace2* expression was not only higher in old mice (Figure 2d) and in females (Figure 2e), but it was also highly heterogenous between individuals (Figure 2f). Using a population conformity comparison (Global *vs*. Eye samples; Figure 2f), we identified two individuals with a significantly lower profile of *Ace2* expression (Eye 2 and Eye 4; Figure 2f), and one with an expression up to 25 times more elevated (Eye 6; Figure 2f). Concerning *Tmprss2*, similar observations were also performed, as its level of expression was significantly dependent on the age and on the sex of the animals; old and female mice both displayed significantly higher levels of *Tmprss2* expression (Figure 2d,e). Moreover, while its global expression profile followed a normal and Gaussian distribution, a heterogenicity of *Tmprss2* expression between individuals was also noticed (Global *vs.* Eye samples; Figure 2f). The accuracy of our samples was finally assessed by sequence analysis (Figure 2g).

Taken together, our *Ace2*/*Tmprss2* gene profiling demonstrates a heterogenicity of expression that is particularly relevant between individuals.

## 4. Discussion

Ocular symptoms associated with SARS-CoV-2 infection are not homogenous between affected individuals. Based on this observation, which we noted within our own laboratory, we initiated a series of experiments to understand and interpret this interindividual discrepancy.

Therefore, we initially focused on the expression of SARS-CoV-2 viral binding sites in the eyes of the mice. We were able to observe that, as with other tissues susceptible to infections by this virus, ACE2 could be co-expressed with the CK18 marker. Interestingly, CK18 is part of the cytoskeleton [58,63] and, therefore, seems to be an integral component of the structures involved in chemodetection and/or biological secretions. Indeed, at the pulmonary level, but also at the sensory level, most CK18-positive cells (e.g., sustantacular and taste cells) have microvilli-like structures [36,39,65,66]. Therefore, we confirmed this correlation in the mouse eye by observing the expression of CK18/ACE2 at the level of the lumen of the intraorbital lacrimal and Harderian glands, which are both involved in the secretion of biological fluids and are composed of microvillar sensor cells [51,63,67,68]. SARS-CoV-2, therefore, seems to target cells that, by their function, are in contact with the environment (chemodetectors and secretary/sensor cells).

By comparing the level of expression of the viral-associated binding sites in the eyes with the ones measured at the olfactory and gustatory levels, we found that it was lower and heterogenous. Therefore, an initial ocular infection of SARS-CoV-2, although possible [10,19,20,21,23], seems less favorable than the one resulting from the upper airway infection [14,15,36,61,69]. Although the RNA expression profiles were also correlated with our histological investigations, a precise quantification of the protein levels in the various mouse ocular subregions still needs to be performed. As such, and by comparing our results of gene and protein expression at the ocular level, we were able to notice that even at a relatively low level of RNA expression, the protein signal of ACE2 was relevant. The apparent discrepancy in the observed signal intensity could imply that ACE2 expression is regulated at both transcriptional and translational levels [70], as well as that it is sensitive to individual and external factors, such as hormonal status and/or environmental conditions [71]. Nevertheless, we identified areas where the expression of ACE2 could be ectopically robust, such as at the level of the RPE layer of the retina, the intraorbital secretion glands (lacrimal and Harderian) and the optic nerve. As such, during a secondary systemic infection via the blood vessels, the nasolacrimal duct or via neurotropic ways, such as the optic nerve, if a parallel expression can be established in humans, the virus could introduce itself, reproduce, and generate cytopathic destruction and inflammatory processes [16,17,18,61,72] at the previously mentioned ocular substructures. This mechanism could, therefore, partially explain the delayed, sex- and age-dependent ocular symptoms observed episodically in humans and mice [7]; these include, respectively, cases of blindness [13,62], dry eye [73,74,75] and optic neuritis with demyelinating lesions [11,12].

In this study, we used mice with the C57BL/6 genetic background as an animal model and we were able to observe ocular heterogenicity in the expression of SARS-CoV-2 binding genes. It could be also interesting to confirm this heterogenicity in the other animal and mouse models commonly used in the laboratory, especially those related to COVID-19 research, such as hamsters, ferrets or BALB/c and K18-hACE2 mice [32,33,76]. Moreover, and for a translational consideration, direct proof of the ocular heterogeneity of the expression of ACE2/TMPRSS2 in humans remains to be verified. Mice *Ace2* and *Tmprss2* are genetically distant from humans and thus display distinct SARS-CoV-2 binding affinity [14,15]; however, it could also undergo an altered regulation of expression [76,77]. Moreover, mice and human eyes also show different anatomical features [78,79], particularly evident in the Harderian glands, which are largely absent in the adult human [68,80]. These species-related particularities must, therefore, be considered when comparing and interpreting ocular symptoms between mice and humans following SARS-CoV-2 infection. Nevertheless, recent studies demonstrate the occurrence of viral load or the identification of the viral particles in human ocular regions, such as in the lacrimal glands, the optic nerve or in the retina [64,81], arguing for a positive correlation of expression between the two species.

Viral transmission from eye tissue and its related secretions thus needs to be seriously considered, particularly regarding the direct proximity between persons or vector animals that could be encountered; this includes, for example, frontline workers, including medical personal or researchers, as well as vulnerable populations [82,83]. Mitigation approaches, in order to prevent direct ocular transmission could, therefore, include wearing eye shields or the application of a blocking agent, such as heparin-based formulations in the exposed face area [84]; these measures would be potent and personal protective approaches to limit the steady rise of viral transmission and ocular symptoms [85].

SARS-CoV-2-related mortality has rapidly decreased with the collective immunity raised by natural past viral exposures or vaccination efforts [86]. However, different levels of viral infectivity/transmissibility, and symptoms related to the emerging SARS-CoV-2 variants, including from the ocular region, could further be encountered [87]. In this study, we did not investigate this issue. Recently, the mouse has emerged as the new potential reservoir host of the SARS-CoV-2 omicron variant [44], a paradigm shift that supports the current scientific effort to better describe the viral strategy taking place in mice. The use and characterization of this animal model is, therefore, fundamental to understanding these mechanisms, not only of its viral susceptibility and infection, but also regarding the development of the associated symptoms and further viral transmission [30,32,88]. In this study, we characterized the expression of *Ace2*/*Tmprss2* viral-associated binding sites in the mouse eye. Based on our observations, we thus propose an interpretation of the heterogenous etiology of the COVID-19 ocular symptoms and provide the scientific community with an ocular mapping of viral-related binding sites, the potential localization of viral entry, and further translational studies.

## Figures and Tables

**Figure 1 viruses-15-00354-f001:**
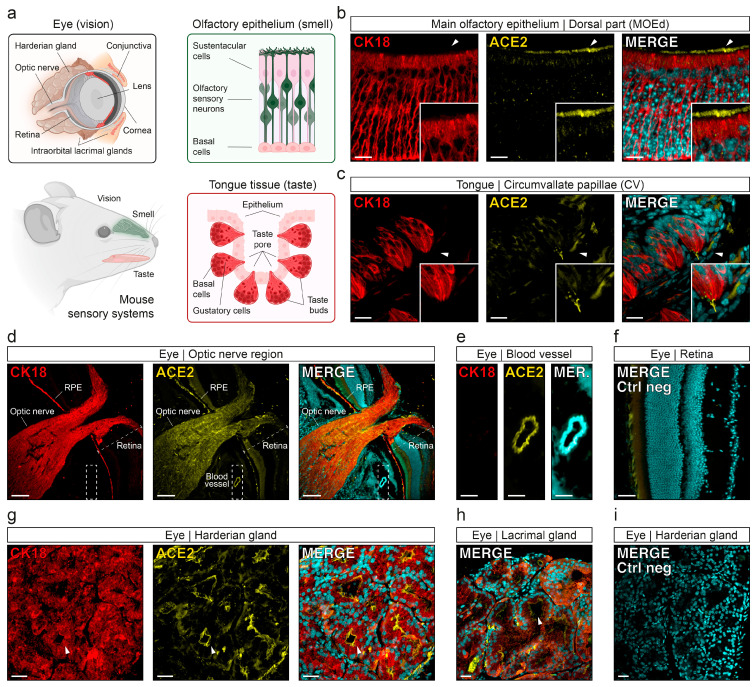
Co-expression of CK18/ACE2 in mouse sensory systems. (**a**) Schematic representation of a mouse head and its sensory organs and cells of interest. Illustrations are created with BioRender.com (accessed on 30 November 2022) (**b**–**i**) Immunohistochemical investigation of CK18 (in red) and ACE2 (in yellow) in the different mouse sensory systems. (**b**) In the MOE_d_, ACE2 is mainly expressed in the apical region of the CK18-positive sustentacular cells. Arrowheads indicate the zoom-in region, highlighting the microvillar region of sustentacular cells. (**c**) In the CV of the tongue, ACE2 is found in the covering cells of the tongue epithelium, in the sensory taste pore and cellular body of CK18-positive gustatory cells. Arrowheads indicate the zoom-in region, highlighting the taste pore microvilli-like region. (**d**–**i**) Identification of CK18-positive subtissular regions of the mouse eye. (**d**) In the optic nerve region, ACE2 is ectopically expressed in the CK18-positive optic nerve and retinal pigment epithelium (RPE). (**e**) ACE2-positive blood vessel (zoom-in view of the dashed white rectangle in (**d**)). (**f**) Negative control of the retina epithelium. (**g**) In Harderian glands, ACE2 is expressed in the lumen of the CK18-positive tubular alveoli (white arrowhead). (**h**) In the intraorbital lacrimal glands, ACE2 can be found in the lumen of the acini cells (white arrowhead). (**i**) Negative control in the Harderian glands. Scale bars are, 10 μm (**b**), 20 μm (**c**), 100 μm (**d**), 25 μm (**e**,**f**), 30 μm (**g**–**i**). Here, representative immunostainings were obtained from OMP-GFP mice, *Gfp*^+/+^ (**b**) and *Gfp*^−/−^ (**c**–**i**). Nuclei are counterstained with Dapi (in cyan (**b**–**i**)).

**Figure 2 viruses-15-00354-f002:**
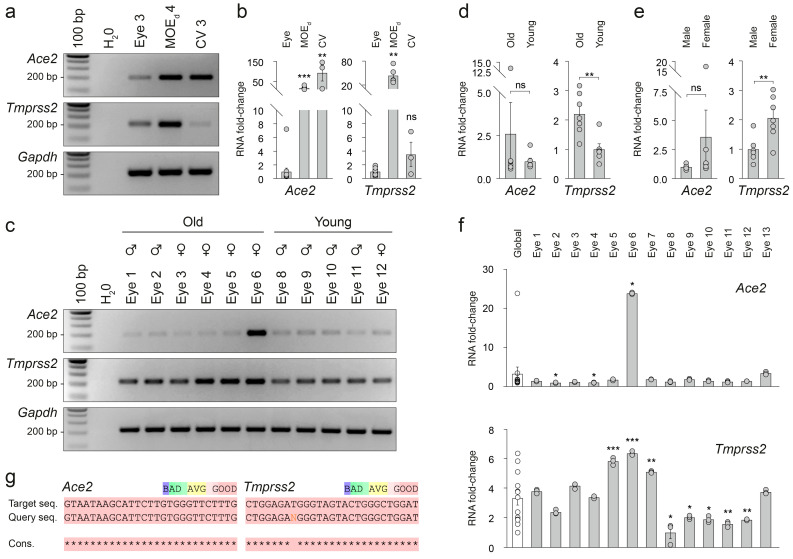
*Ace2*/*Tmprss2* gene expression profile in the mouse eye. (**a**) Representative RT-PCR results comparing *Ace2*/*Tmprss2* expression between the mouse sensory systems (visual, olfactory and taste systems). For the MOE_d_ 4 sample, GFP-expressing MOE_d_ from *Gfp*^+/− and +/+^ mice were used. (**b**) Quantification by RT-qPCR of *Ace2*/*Tmprss2* expression between mouse sensory systems. (**c**) Representative RT-PCR result comparing *Ace2*/*Tmprss2* expression between male (♂), female (♀), old and young mouse eye samples obtained (from Table 1), illustrating individual and ectopic expression of SARS-CoV-2-associated binding genes. (**d**–**f**) Quantification by RT-qPCR of age, sex and interindividual differences between eye samples (Table 1). (**d**) Age-dependent expression of *Ace2*/*Tmprss2* genes in the mouse eye. (**e**) Sex-dependent expression of *Ace2*/*Tmprss2* genes in mouse eyes. (**f**) Conformity analysis between samples and global population (Global, in white) reveals the interindividual heterogenicity of *Ace2*/*Tmprss2* gene expression in the mouse eye. (**g**) Representative assessment of our fragment specificity, here with the Eye 6 sample. The obtained sample sequences (Query seq.) are aligned with their respective *Ace2* or *Tmprss2* target sequences (Target seq.). In case of a non-determined nucleotide, the symbol “N” is used. The level of the sequence conservation (Cons.) is indicated by asterisks and the color code scale. (**a**–**f**) H_2_O is used as a negative control of transcript expression and *Gapdh* as a reporter gene. Data are expressed as an RNA fold change and represented as mean ± SEM with aligned dot plots for *n* ≧ 3 samples/replicates. Comparisons between conditions and/or samples are performed with two-tailed Student’s *t*-tests or Mann–Whitney tests, * *p* < 0.05, ** *p* < 0.01, *** *p* < 0.001, ns for non-significant. Ladder of 100 base pairs (bp, (**a**,**c**)).

**Table 1 viruses-15-00354-t001:** Sample characteristics used in RT-PCR and RT-qPCR analysis.

Sample ^1^	Mouse Age (Weeks)	Mouse Sex
Eye 1 (1× eye)	65	Male
Eye 2 (1× eye)	65	Male
Eye 3 (1× eye)	65	Female
Eye 4 (1× eye)	65	Female
Eye 5 (1× eye)	65	Female
Eye 6 (1× eye)	65	Female
Eye 7 (1× eye)	65	Female
Eye 8 (1× eye)	3	Male
Eye 9 (1× eye)	3	Male
Eye 10 (1× eye)	3	Male
Eye 11 (1× eye)	3	Male
Eye 12 (1× eye)	3	Female
Eye 13 (1× eye)	3	Female
Eye 14 ^2^ (1× eye)	3	Male
MOE_d_ 1 (2× MOE_d_)	65	Male (1×) + Female (1×)
MOE_d_ 2 (2× MOE_d_)	65	Male (1×) + Female (1×)
MOE_d_ 3 (3× MOE_d_)	80	Male (1×) + Female (2×)
MOE_d_ 4 (2× MOE_d_)	65	Male (1×) + Female (1×)
MOE_d_ 5 (3× MOE_d_)	60	Male (2×) + Female (1×)
CV 1 (2× CV)	65	Male (1×) + Female (1×)
CV 2 (3× CV)	60	Male (2×) + Female (1×)
CV 3 (3× CV)	65	Male (1×) + Female (2×)

^1^ For Eye and CV samples, OMP-GFP mice with no olfactory-related GFP expression (*Gfp*^−/−^) were used. For MOE_d_ samples, OMP-GFP mice with olfactory-related GFP expression (*Gfp*^+/− and +/+^) were used. ^2^ RT-qPCR analysis from Eye 14 sample revealed an abnormally elevated C_T_ for *Gapdh* and RNA extractions from this sample were accordingly not used for further analysis.

## Data Availability

The data presented in this study are available upon request to the corresponding authors.

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
