# Peer review of "Ocular Symptoms Associated with COVID-19 Are Correlated with the Expression Profile of Mouse SARS-CoV-2 Binding Sites"

_viruses, 2023, doi:10.3390/v15020354_

Round 1

Reviewer 1 Report

The authors characterized the expression of ACE2/TMPRSS2 in the eyes from wild type C57BL/6 mice (Mus musculus; Janvier Labs), heterozygous and homozygous OMP-GFP mice, of both sexes, using histological and RT-PCR based methods. This study provided interesting evidences that ACE2 was ectopically expressed in subtissular ocular regions such as in the optic nerve and in the Harderian/intraorbital lacrimal glands, and also reported that the expression of Ace2/Tmprss2 varied between the age and sex of the animal as well as heterogenous between individuals, which might help to interpret the human ocular-associated symptoms linked to SARS-CoV-2. However, the authors should further explain the following issues.

1.    Although the binding of the spike (S) protein to its receptor, angiotensin-converting enzyme 2 (ACE2), and subsequent membrane fusion play a very key role in the mechanisms of SARS-CoV-2 entry into host cells, ACE2 is not equal to SARS-CoV-2 viral entry sites. The author has many such descriptions in this manuscript, which should be modified accordingly, especially in the section of results. The author did not conduct SARS-CoV-2 related research and should not involve virus related statements in the section of results, which could be discussed in the section of Discussion.

2.    The author stated an heterogenicity of expression of ACE2/Tmprss2 in-between animal individuals, which should be pointed out more clearly that these differences were found in wild type, heterozygous or homozygous animal models? What might be the influencing factors.

3.    Fig2C showed a very significant different RNA levels of ACE2 expression in animal models. It is well known that RT-PCR and SYBR® Green enzyme based qRT-PCR are very easy to be contaminated, which affects the judgment of results. Whether these amplified fragments had been sequenced to determine the specificity of amplification.

Author Response

Responses to Reviewer #1

We would like to thank the Reviewer for her/his time and precise review. We really appreciated her/his relevant comments and we think that the revision based on her/his requests will definitely strengthen our manuscript and allow a better understanding and interpretation of our results. We have now submitted a new manuscript in which we have highlighted all changes in blue.

Please find below our point-by-point responses:

  1. Although the binding of the spike (S) protein to its receptor, angiotensin-converting enzyme 2 (ACE2), and subsequent membrane fusion play a very key role in the mechanisms of SARS-CoV-2 entry into host cells, ACE2 is not equal to SARS-CoV-2 viral entry sites. The author has many such descriptions in this manuscript, which should be modified accordingly, especially in the section of results. The author did not conduct SARS-CoV-2 related research and should not involve virus related statements in the section of results, which could be discussed in the section of Discussion.

We agree with the Reviewer and thank her/him for her/his relevant comments. As we indeed have no direct experimental evidence of viral entry into the ocular sites that we have studied, we have now systematically changed “SARS-CoV-2 entry sites” into " SARS-CoV-2 binding sites" in the title, abstract and main text. As suggested by the reviewer, we have now discussed this important point in the discussion section.

  1. The author stated an heterogenicity of expression of ACE2/Tmprss2 in-between animal individuals, which should be pointed out more clearly that these differences were found in wild type, heterozygous or homozygous animal models? What might be the influencing factors.

We would like to thank the reviewer for this relevant comment. We apologize for the lack of clarity of our initial “Animals” description in the Materials and Methods section. As suggested by the reviewer, we have now described more clearly the genotypes used for comparisons in our methods, Table 1 and result sections. In summary, the mice used in this study all came from a single mouse line, the "OMP-GFP" line built on a C57BL/6 genetic background. In this mouse line, GFP is under the control of the Omp promoter (exclusively expressed in olfactory neurons), consequently making the olfactory neurons GFP+ in homozygous (Gfp+/+) and heterozygous (Gfp+/-) mice. This fluorescence is therefore only exploited in order to make the precise sampling of olfactory tissue (MOEd). Thus, for the eyes and the CV, mice littermates not expressing GFP (Gfp-/-) were preferred (mice that we called inappropriately “Wild type” during our initial description). It should be noted that the expression of GFP would in no way affect the expression of ACE2/TMPRSS2 in the MOEd (Brechbühl et al., Commun Biol, 2021).

Our observations regarding the individual heterogenicity of ACE2/TMPRSS2 expression are therefore not related to a particular genotype but may reflect other published intrinsic and extrinsic factors such as hormonal status or environmental conditions that we have now mentioned in our discussion. We also mentioned in the discussion that other animal or mice models (BALB/c, K18-hACE2, …) should now be checked for this aspect.

  1. Fig2C showed a very significant different RNA levels of ACE2 expression in animal models. It is well known that RT-PCR and SYBR® Green enzyme based qRT-PCR are very easy to be contaminated, which affects the judgment of results. Whether these amplified fragments had been sequenced to determine the specificity of amplification.

We thank the reviewer for this important methodological question. In our experiments, we systematically used DNAse and primers that span exon-exon junctions to avoid any potential endogenous contamination/amplification of genomic DNA (Brechbühl et al., Commun Biol, 2021). Moreover, independent experiments (from our initial ARN purifications samples) were performed with reproducible results (from cDNA conversion to RT-qPCR observations). We have now described more precisely our results in the revised manuscript.

In addition, the melting temperature (Tm) profiles were systematically analyzed and always revealed, for each run performed (sample, gene), a single maximal peak consistent with a pure and unique amplification. To complement these observations, we have now analyzed the qPCR products by electrophoresis and sequencing. We have modified our Figure 2 and added a new Supplementary Figure S1, both displaying this important notion. We have modified the text accordingly.

Reviewer 2 Report

This is significant research pertinent to current precautions against SARS-CoV-2. Faceguards were an essential PPE for first-line workers.

Transmission from the eyes is a known fact Authors need to discuss translational aspects of discussing mitigating measures to avoid ocular transmission e.g face shields or other agents (10.1016/j.ijbiomac.2021.04.148). While mortality has rapidly gone down with the vaccination and past exposures/infections. There is a steadier rise in peculiar symptoms which are severe and often confounding. 

All in all, this is a well planned and executed study.

Author Response

Response to Reviewer #2

We would like to thank the Reviewer for her/his time and precise review. We really appreciated her/his relevant comment and we think that the revision based on her/his request will definitely strengthen our manuscript and allow a better understanding and interpretation of our results. We have now submitted a new manuscript in which we have highlighted all changes in blue.

Please find below our point-by-point responses:

  1. Transmission from the eyes is a known fact Authors need to discuss translational aspects of discussing mitigating measures to avoid ocular transmission e.g face shields or other agents (10.1016/j.ijbiomac.2021.04.148). While mortality has rapidly gone down with the vaccination and past exposures/infections. There is a steadier rise in peculiar symptoms which are severe and often confounding.

We would like to thank the Reviewer for her/his encouraging remarks and relevant suggestion. We have now added these important medical and translational notions with the corresponding references in our discussion section.

Reviewer 3 Report

In this manuscript, Brechbühl et al. evaluated the expression profiles of SARS-CoV-2 viral entry receptors, including ACE2 and TMPRSS2, in the ocular system in mice, with the purpose to find out the correlation between their expression and the related COVID symptoms. The study was explained clearly, and the manuscript can be considered for publication if the authors could address the following comments.

The authors here utilized mouse as the animal model for studying the ACE2 and TMPRSS2 expression profile, however, there are quite some differences between the systems in mouse and human. The authors may need to emphasize this gap between species in the discussion section if no other supporting experiments could be added.

For the immunostaining experiment, how did the authors normalize the intensity of the stains from different samples? In line 215 the authors indicated there were 6 eyes from 5 different mice, so how do the staining of two eyes from the same mouse compare? In addition, from figure 1 it seems that the ocular system has decent level of staining signals of ACE2, however, in figure 2 the RT-PCR results showed significantly lower expression of ACE2 in eyes, could the authors mention this difference?

The authors may include one or more sentences in the introduction to mention some examples of other viruses that infect the ocular system.

In line 55-56, the authors mentioned that the COVID symptoms are depending on age or sex as well as the different emerging variants. However, in the following study described the authors didn’t look at the potential different effects by different variants, so it would be better if the authors could mention that aspect.

There are also some places where the language needs to be modified. For example, in line 186, the word “corelated” should be “correlated”. And in line 71-72, the sentence “Interestingly and demonstrating the abilities of SARS-CoV-2 to rapidly evolve and 71 adapt to the environmental pressure…” may need to be re-written.

Author Response

Responses to Reviewer #3

We would like to thank the Reviewer for her/his time and precise review. We really appreciated her/his relevant comments and we think that the revision based on her/his requests will definitely strengthen our manuscript and allow a better understanding and interpretation of our results. We have now submitted a new manuscript in which we have highlighted all changes in blue.

Please find below our point-by-point responses:

  1. The authors here utilized mouse as the animal model for studying the ACE2 and TMPRSS2 expression profile, however, there are quite some differences between the systems in mouse and human. The authors may need to emphasize this gap between species in the discussion section if no other supporting experiments could be added.

We agree with the Reviewer and thank her/him for her/his relevant comments. We have now discussed this important notion in our discussion section and commented on the comparison of ocular ACE2 and TMPRSS2 expressions between mice and humans.

  1. For the immunostaining experiment, how did the authors normalize the intensity of the stains from different samples? In line 215 the authors indicated there were 6 eyes from 5 different mice, so how do the staining of two eyes from the same mouse compare? In addition, from figure 1 it seems that the ocular system has decent level of staining signals of ACE2, however, in figure 2 the RT-PCR results showed significantly lower expression of ACE2 in eyes, could the authors mention this difference?

We thank the Reviewer for her/his relevant methodological question and we apologize for the lack of clarity in the original description of our experimental design. We have now modified our method section to explain that our confocal acquisitions were performed under a calibrated intensity of the lasers in order to standardize the observations and the comparisons of the signals emitted within each sensory tissue investigated.

Concerning the results from the two eyes obtained from the same mouse, similar signals were observed. Since the results between the left eye and the right eye came for a single mouse, it does not allow us to draw any conclusion and we therefore prefer not to mention it in our description of the results.

Concerning the different levels of ACE2 signals obtained by immunostainings and the one obtained by qPCR, this apparent discrepancy could be partially explained by the fact that ACE2 expression is known to be regulated at both transcriptional and translational levels (Hu et al., 2021) as well as sensitive to individual and external factors such as hormonal status and/or environmental conditions (Rath et al., 2022). As recommended by the reviewer, we have now mentioned this difference in our discussion section.

  1. The authors may include one or more sentences in the introduction to mention some examples of other viruses that infect the ocular system.

We thank the Reviewer for her/his suggestion. We have now included examples of ocular viral infections in our introduction.

  1. In line 55-56, the authors mentioned that the COVID symptoms are depending on age or sex as well as the different emerging variants. However, in the following study described the authors didn’t look at the potential different effects by different variants, so it would be better if the authors could mention that aspect.

We thank the Reviewer for her/his remark. We have now mentioned the potential variant effects in our discussion and added related references. We have also mentioned the fact that we did not address this issue experimentally in this study.

  1. There are also some places where the language needs to be modified. For example, in line 186, the word “corelated” should be “correlated”. And in line 71-72, the sentence “Interestingly and demonstrating the abilities of SARS-CoV-2 to rapidly evolve and 71 adapt to the environmental pressure…” may need to be re-written.

We apologize for the mistakes and have now carefully proofread the whole manuscript in order to improve the grammar, the spelling and the English.

Round 2

Reviewer 1 Report

The authors have made corresponding modifications according to the comments. The conclusion is reasonable and helpful to understand the symptoms and signs of the ocular system after human infection with SARS-CoV-2.